# Stabilizing the Oil-in-Water Emulsions Using the Mixtures of *Dendrobium Officinale* Polysaccharides and Gum Arabic or Propylene Glycol Alginate

**DOI:** 10.3390/molecules25030759

**Published:** 2020-02-10

**Authors:** Bo Wang, Haiyan Tian, Dong Xiang

**Affiliations:** 1College of Food Science and Engineering, Hainan University, Haikou 570228, China; 15607662867@163.com (B.W.); Tianhy2019@163.com (H.T.); 2Engineering Research Center of Utilization of Tropical polysaccharide resources, Ministry of Education, Haikou 570228, China

**Keywords:** *Dendrobium officinale* polysaccharides, propylene glycol alginate, gum arabic, coconut oil-in-water emulsion

## Abstract

Coconut oil-in-water emulsions were prepared using three polysaccharides: *Dendrobium officinale* polysaccharide (DOP), propylene glycol alginate (PGA), gum arabic (GA) and their polysaccharide complexes as emulsifiers. The effects of the ratio of the compounded polysaccharides on their apparent viscosity and interfacial activity were explored in this study. The average particle size, zeta potential, microstructure, rheological properties, and physical stability of the emulsions prepared with different compound-polysaccharides were studied. The results showed that mainly DOP contributed to the apparent viscosity of the compound-polysaccharide, while the interfacial activity and zeta potential were mainly influenced by PGA or GA. Emulsions prepared with compound-polysaccharides exhibited smaller average particle sizes, and microscopic observations showed smaller droplets and less droplet aggregation. In addition, the stability analysis of emulsions by a dispersion analyzer LUMiSizer showed that the emulsion prepared by compounding polysaccharides had better physical stability. Finally, all of the above experimental results showed that the emulsions prepared by PGA:DOP = 2:8 (total concentration = 1.5 wt%) and 2.0% GA + 1.5% DOP were the most stable.

## 1. Introduction

Emulsions are applied in various fields including food, cosmetics, pharmaceuticals, petrochemicals, and agricultural products [1,2,3,4]. An emulsion is a thermodynamically unstable system of two immiscible liquids. Emulsions are categorized into two types: oil-in-water (O/W) and water-in-oil (W/O) emulsions [1,5]. Oil-in-water emulsions are extensively used in food products including milk, beverages, sauces, etc., and oil-in-water emulsions are utilized for delivering lipophilic bioactive compounds such as vitamins and antioxidants [1,6,7,8]. The stability of the emulsions is limited by different phenomena including coalescence, flocculation, lactation, and Ostwald ripening [2,3,4,5,7]. The stability of the emulsions can be improved by adding emulsifiers and stabilizers. Polysaccharides are often used as stabilizers and thickeners in the food industry [2,9]. Polysaccharides, when added to oil-in-water emulsions, increase the viscosity of the aqueous phase, enhance the spatial repulsive force and electrostatic repulsion between the oil droplets, and alter the rheological properties of the emulsion [3,5,8,9,10]. In recent years, scientists have studied the emulsification properties of many natural polysaccharides such as seaweed polysaccharides [10], pectin [8], spruce galactoglucomannans [11], orchid-root polysaccharides [12], basil-seed polysaccharides [13], etc.

*Dendrobium officinale*, a precious Chinese herbal plant, is distributed mainly in tropical and subtropical regions [14,15]. Polysaccharides from *Dendrobium officinale* have demonstrated many biological activities including antioxidant, antitumor, hypoglycemic, and immunomodulatory activities [15,16,17,18,19,20,21,22,23]. Previous studies have shown that Dendrobium candidum polysaccharide is mainly composed of mannose and glucose, and its molecular weight is between 178 KDa and 394 KDa [14,18,22]. Unfortunately, no research has been conducted on the emulsification properties of *Dendrobium officinale* polysaccharides. Propylene glycol alginate (PGA) is a derivative of propylene oxide and alginic acid; PGA is a linear polysaccharide with 50–85% of esterified carboxyl groups [24,25,26]. PGA is widely used in the food industry as a viscosity enhancer and stabilizer [27,28,29,30]. Gum arabic (GA), a natural complex polysaccharide with good emulsifying properties and high water-solubility, is widely used as an emulsifier in the food industry [31,32]. Gum arabic is composed of arabinogalactan (AG), arabinogalactan protein (AGP), and glycoprotein (GP). The emulsifying property of GA is majorly contributed by AGP [33,34,35,36,37].

PGA contains numerous esterified carboxyl groups, but its hydrophilicity is weak. DOP is a highly hydrophilic natural polysaccharide, so the compound polysaccharide of DOP and PGA is likely to be a good emulsifier. Additionally, GA possesses high hydrophilicity and low viscosity. As an emulsifier, a high quantity of GA is required to prepare a stable emulsion [38,39]. However, DOP can greatly increase the viscosity of the continuous phase, so the complex polysaccharides of GA and DOP can exert good emulsifying effects. In this study, an oil-in-water emulsion was prepared using a compound of *Dendrobium officinale* polysaccharide, GA or PGA as an emulsifier, and coconut oil as an internal dispersed phase. The apparent viscosity and interfacial tension of the mixed polysaccharides were investigated, and the stability of the water–oil emulsions was studied. The emulsified concentration of the compound polysaccharide was investigated to prepare a stable emulsion. The particle size distribution, zeta potential, microstructure, and rheological properties of the oil-in-water emulsion were measured. This study provides a better application of *Dendrobium officinale* polysaccharides as an emulsifier in natural functional emulsions.

## 2. Results and Discussion

### 2.1. Apparent Viscosity of Polysaccharides

In Figure 1A, the apparent viscosity of PGA and DOP increased while increasing the concentration, which is a common feature of macromolecular polysaccharide aqueous solutions. When the concentration of the polysaccharide molecule was increased, the interaction between the polysaccharide molecules increased, which resulted in an increase in the viscosity of the polysaccharide solution [40]. However, no significant change has been observed in the viscosity of the aqueous solution of GA at a concentration of 0.1–1%, because GA has good water solubility and low viscosity [32], and a significant increase in the viscosity can be observed only at a concentration of 10% or more. In addition, Figure 1B shows the effect of the ratio of the compound-polysaccharide (total concentration was 1%) on its viscosity. The viscosity of the compound-polysaccharide was closely related to the ratio of DOP and PGA or the ratio of DOP and GA. Additionally, the content of DOP could significantly affect the viscosity of the two polysaccharides. For PGA:DOP = 2:8 and GA DOP = 2:8, the viscosities of the two polysaccharides were the largest. These results indicated that the apparent viscosity of the polysaccharides in aqueous solutions was primarily provided by DOP.

### 2.2. Interfacial Tension of the Polysaccharides

The interfacial properties of the emulsifier reflect its emulsion stabilizing property [10]. In Figure 2A, compared to the interfacial tension between coconut oil and water (23.46 mN/m) [6,41], the interfacial tension of the three polysaccharide aqueous solutions was significantly reduced. The interfacial tension of the polysaccharide solution decreased while increasing the concentration of the polysaccharide; previous studies that reported that the polysaccharide molecules were adsorbed on the oil–water interface agreed with the above observation [6,42,43]. Interestingly, the interfacial tension of PGA was the lowest at a concentration of 0.25%, and the interfacial tension of DOP was the lowest at a concentration of 0.75%, but the interfacial tension of GA continued to decrease within a concentration range of 0.1–1%. This phenomenon indicates that the adsorption amount at the oil–water interface gradually increases while increasing the concentration of the emulsifier, and when the adsorption amount is saturated, it reaches critical micellar concentration (CMC); at this time, the interfacial tension reaches a steady-state [44,45]. Furthermore, the ability of GA to reduce the interfacial tension was the best of the three polysaccharide forces, which was due to the low molecular weight of GA and the proteins in GA. Compared to large molecular polysaccharides, small molecular polysaccharides and surface hydrophobic proteins could be adsorbed to the oil–water interface faster [38,43]. In Figure 2B, the interfacial activity of the compound-polysaccharides mainly depended upon the contents of GA and PGA, which was the opposite to their apparent viscosities. The surface hydrophobicity of the compound-polysaccharides was mainly provided by PGA and GA, and for PGA:DOP = 8:2 and GA:DOP = 8:2, the compound polysaccharide exhibited the smallest interfacial tension. It is interesting, however, that when PGA:DOP = 5:5 and GA:DOP = 4:6, their interfacial tension was the largest. This shows that the value of interfacial tension does not change regularly with the proportion of PGA or GA in the complex polysaccharide. This might be when PGA:DOP or GA:DOP was in a certain ratio; the emulsifier molecules interacted with each other and led to reduced group exposure.

### 2.3. The Particle Size Distribution of the Emulsions

The particle size distribution of the emulsion can reflect the droplet flocculation and aggregation [43]. In Figure 3, the particle size distribution curves of all emulsions were single-peaked, which indicated that all emulsions formed uniform droplets. The emulsion prepared by PGA alone exhibited the largest average particle size. As described in Section 3.2, PGA displayed the weakest ability to reduce interfacial tension, and it was not effectively adsorbed on the droplet surface [46]. In contrast, the average particle size of the emulsion prepared by GA alone was smaller than the PGA due to the excellent adsorption rate of GA [37]. However, the particle size of the emulsion prepared by DOP alone was the smallest of the three polysaccharides, which might benefit from the high viscosity of DOP, the limited movement of the droplets, and re-agglomeration [2]. When PGA was mixed with DOP, the average particle size of the emulsion was significantly reduced. For PGA:DOP = 2:8, the average particle size was the smallest (8.356 ± 0.061 μm). Meanwhile, for GA:DOP = 2:8, the average particle size was 11.768 ± 0.015 μm. This result validates our conjecture that DOP was highly hydrophilic, while PGA and GA contained hydrophobic groups. When DOP was mixed with PGA or GA, a mixed emulsifier containing both hydrophilic and hydrophobic groups was formed [41]. This mixed emulsifier easily formed stable emulsions.

### 2.4. Zeta Potential

Zeta potential characterizes the surface charge of the droplets and reflects the repulsive force between the emulsion droplets [47]. According to Figure 4, the zeta potential of all emulsions was negative, and the zeta potential value was quite high. Previous studies have reported that both PGA and GA contain negative charges [37,41,42]. The emulsion prepared by DOP alone exhibited the smallest amount of surface charge, while the amount of charge of the emulsion prepared by compounding polysaccharides depended upon the ratio of PGA or GA. For PGA:DOP = 8:2, the emulsion displayed the largest zeta potential value, but for GA:DOP = 8:2, the zeta potential value of the emulsion was second only to that of the emulsion prepared by GA alone. This phenomenon showed that the compound polysaccharides could be used as emulsifiers to provide better electrostatic repulsion between emulsion droplets.

### 2.5. Fluorescent Microstructure of Emulsions

Figure 5 shows the microstructure of the emulsions under a fluorescence microscope. The emulsion prepared by 1% GA alone and the emulsion prepared by 1% PGA contained larger droplets and large amounts of aggregation, which agreed with their average particle size in Section 3.3. However, compared to the emulsion prepared by 1% GA alone or 1% PGA alone, the emulsion prepared by 1% DOP alone displayed more small-sized droplets and less aggregation. Two main types of repulsions prevent the aggregation of emulsion droplets: steric and electrostatic repulsion [38,48]. Polysaccharides can increase the viscosity of the aqueous phase and strengthen the steric repulsion of the emulsion [49]. The emulsion stabilized by DOP alone possessed the smallest zeta potential, but small droplets (from Section 3.4), indicating that the high viscosity of the DOP aqueous phase prevented the aggregation of the droplets. For the emulsion prepared by the compound polysaccharides, the microstructure of emulsions indicated that the droplet size and degree of aggregation of the emulsion were highly depended upon the ratio of PGA:DOP or GA:DOP. The higher the proportion of PGA or GA in the compound polysaccharide, the larger the emulsion droplets, and the more severe the aggregation of the emulsion droplets, suggesting that the viscosity was a crucial factor for maintaining the stability of the emulsion under this condition.

### 2.6. Visual Observation of Emulsions

The visual observations of all emulsions are shown in Figure 6. The results of the visual observation were consistent with the results of the particle size distribution and microstructure. After four days of storage, significant phase separation occurred in all emulsions except those two emulsions prepared with PGA:DOP = 2:8 (1 wt%) or GA:DOP = 2:8 (1 wt%) as emulsifiers. After six days of storage, the two remaining emulsions also displayed phase separation. According to the above results, the repulsive effect in emulsions could not prevent the droplets from agglomerating during storage. Therefore, we adjusted the concentration and ratio of the compound polysaccharide emulsifier in our further research.

### 2.7. Apparent Viscosity, Zeta Potential and Particle Size Distribution of the Emulsions

According to Figure 7A, the apparent viscosity and zeta potential of the emulsion changed significantly after adjusting the concentration and ratio of the compounded polysaccharide emulsifier. In emulsions prepared by mixing polysaccharides (GA and DOP), GA primarily contributed to the surface charge, and the amount of the surface charge increased while increasing the concentration of GA. However, when the concentration of GA was gradually increased to 2%, the viscosity of the emulsion reached the maximum value; when the concentration of GA was increased to 5%, the viscosity of the emulsion decreased to a minimum. The viscosity of the polysaccharide system was mainly contributed by DOP, while the viscosity of the emulsion system mainly depended upon the viscosity of the continuous phase [50]. When the GA content in the system gradually increased, DOP no longer dominated, and the viscosity of the emulsion decreased. In addition, many GA molecules in solution might help disperse the DOP molecules more and weaken the interaction between the DOP molecules. Additionally, as the content of GA increased, many GA molecules replaced the adsorbed DOP on the droplet surface due to competitive adsorption, which weakened the interaction between the droplets and reduced the viscosity [35,51]. The particle size distribution of all emulsions was single-peaked, and all emulsions exhibited small average particle sizes, which indicated that the droplet size of the emulsion could be significantly reduced by adjusting the concentration and ratio of the mixed emulsifier. Among them, the average particle size of the emulsion prepared by PGA:DOP = 2:8 (total concentration = 1.5 wt%) was the smallest, followed by the emulsion prepared by 2.0% GA and 1.5% DOP.

### 2.8. Microstructure and Visual Observation

The microstructure and visual observation of the emulsions (Figure 8) were consistent with the particle size of the emulsions. The microstructure of all emulsions displayed small droplets, but the emulsions prepared by 4.0% GA + 1.5% DOP and the emulsions prepared by 5.0% GA + 1.5% DOP displayed many aggregated droplets, which was related to their lower viscosity. Additionally, extremely high concentrations of the emulsifier might flocculate numerous free (not adsorbed on the droplet surface) emulsifier molecules [38,48]. After seven days of storage, the emulsions prepared by PGA:DOP = 2:8 (total concentration = 1.5 wt%) and those prepared by 2.0% GA+1.5%DOP displayed good stability without phase separation.

### 2.9. Rheological Properties of the Emulsions

Figure 9A displays the steady-state flow curve of emulsions. All of the emulsions exhibited the typical non-Newtonian behavior of shear-thinning in the shear rate range of 1 s^−1^ to 1000 s^−1^. At the same shear rate, the regularity of the values of the viscosity of different emulsions was the same as the result in Section 3.7. We speculated that there might be an interaction between GA and DOP molecules. As the content of GA continues to increase, GA occupied the binding sites on the DOP molecular chain, resulting in a decrease in the interaction between DOP molecular chains and a decrease in viscosity [35]. The properties of the emulsion system depended mainly upon the characteristics of the continuous phase [50]; so, the rheological behavior of the emulsions was attributed mainly due to the mixed polysaccharides in aqueous solutions. When the shear rate was increased, the entanglements between the polysaccharide-chains were disrupted, and the molecular chains were more randomly orientated. Finally, the interactions between the adjacent chains were reduced, and the viscosity was decreased [52,53,54]. Figure 9B shows the linear viscoelastic region of the two emulsions: the emulsion prepared by PGA:DOP = 2:8 (total concentration = 1.5 wt%) and the emulsion prepared by 2.0% GA + 1.5% DOP. The two emulsions displayed a relatively wide linear viscoelastic region, and G′′ was dominant in the entire test strain range; therefore, a 1% strain was selected for frequency-dependent testing. In Figure 9C, G′ and G′′ of the two emulsions increased while increasing the frequency. In the frequency range of 0.1 Hz–10 Hz, G′′ always dominated (i.e., both emulsions displayed viscous-dominated fluid-like behavior) [55,56,57].

### 2.10. Physical Stability of the Emulsions

The physical stability analysis was a space-related and time-related transmission profile over the sample length and the greater change in the transmission with centrifugation, the worse stability of the emulsion [27,58]. In Figure 10A, the transmission curves of all emulsions showed a floating phenomenon (creaming). The transmission curves of the emulsions prepared by 4.0% GA + 1.5% DOP and 5.0% GA + 1.5% DOP were changed significantly, and the transmittance of the two emulsion samples finally exceeded 90%. The transmission curves of the emulsions prepared by PGA:DOP = 2:8 (total concentration = 1.5 wt%) and those prepared by 2.0% GA +1.5% DOP did not change substantially during the test, which indicated that the two emulsions possessed high physical stability. The analytical range of the sample was selected from the meniscus to the bottom to calculate the instability index of the emulsion samples. The speed index of the interface tracking was calculated during the creaming process for a threshold of 16%. In Figure 10B, the emulsions prepared by PGA:DOP = 2:8 (total concentration = 1.5 wt%) and those prepared by 2.0% GA + 1.5% DOP exhibited a lower instability index and smaller speed of interface movement, compared to other emulsion samples. This result was closely related to the viscosity of the emulsion. The high-viscosity continuous phase limited the aggregation of droplets and the movement of molecules. The molecular movement resistance was large, the speed was slow, and macroscopically, the interface movement speed was small.

## 3. Materials and Methods

### 3.1. Materials and Reagents

*Dendrobium officinale* stems were procured from Bao Yuan Tang Company (Hainan, China). The *D. officinale* stem was cultivated in Qiong Zhong County, Hainan Province and harvested after one year.

Gum arabic (GA) was purchased from Aladdin Biochemical Technology Co. Ltd. (Shanghai, China). Propylene glycol alginate (PGA) was procured from Yuanye Bio-Technology Co. Ltd. (Shanghai, China). All other reagents used were of analytical grade.

### 3.2. Extraction of Dendrobium officinale Polysaccharides

Fifty grams of fresh *D. officinale* stems were cut into small pieces (sizes of about 2–3 cm) and then soaked in boiling water for 5 min. Subsequently, the stems were removed from boiling water, and water droplets were removed from the surface. Then, the stems were placed in a crusher, and 1000 mL of deionized water was added into it. The stems were crushed for 3 min and then filtered. The filtrate was centrifuged (10,000× *g*, 30 min), and the supernatant was collected. The supernatant was precipitated with four volumes of 95% ethanol for 24 h. After centrifuging (10,000× *g*, 20 min), the precipitate was freeze-dried to obtain *Dendrobium officinale* polysaccharides (HEPDOP).

### 3.3. Apparent Viscosity Measurement

Apparent viscosity was measured using a Brookfield viscometer (DV3TLVTJ0, Middleboro, MA, USA). A suitable rotor was selected to ensure that the torque was between 10% and 100%. Each sample was measured three times.

### 3.4. Interfacial Activity of Compound Polysaccharides

According to the method of Li, Yujie et al., the drop shape analysis instrument (DropMeter A-60, MAIST, Ningbo, China) was used for determining the interfacial tension between the polysaccharide solution and coconut oil [41]. Each sample was measured three times.

### 3.5. Preparation of Oil-in-Water Emulsion

The polysaccharide solution was hydrated at room temperature overnight, then 5% coconut oil was added and homogenized using a high shear homogenizer (FJ-200; Biaoben Instruments, Shanghai, China) at a speed of 10,000 rpm for 2 min [59].

### 3.6. Droplet Size Distribution of the Emulsions

The average particle diameter and particle size distribution of the emulsions were measured using a laser particle size analyzer (Ineas Physical Optics Instrument Co. Ltd., WJ-60, Shanghai, China). The emulsion was mixed evenly, poured into a stirring tank filled with deionized water, and measured when the shading ratio was 1.5. Each sample was measured three times.

### 3.7. Zeta Potential Measurement

The zeta-potential of the emulsions was measured using particle electrophoresis (Zetasizer Nano S90, Malvern Instruments, Worcestershire, UK), and each sample was measured three times.

### 3.8. Microstructure of Emulsions

The microstructure of the emulsions was observed using an inverted fluorescence microscope (AE31AEFL-1NV, Motic, Hong Kong, China). Coconut oil was colored with 0.1% Nile Red [60], and then emulsions were prepared using the colored oil for observing under a 10× eye lens and a 10× objective lens.

### 3.9. Visual Observation of Emulsions

The prepared emulsion was poured into a 5-mL sample bottle, and the phase separation of the emulsion was captured in a photograph.

### 3.10. The Rheological Behavior of the Emulsions

The rheological properties of the emulsions were measured using a HAAKE Modular Advanced Rheometer System (MARS 40, Thermo Scientific, Waltham, MA, USA). The 35-mm parallel plate was used with a measurement gap of 1 mm. The sample was evenly loaded between the sample pan and the plate, and the viscosity of the emulsion was measured at a shear rate of 1 to 1000 s^−1^ to plot a steady-state curve of the emulsion. The linear viscoelastic region of the emulsion was determined in the strain mode, keeping the strain range between 0.5–500%. The dynamic oscillation test of the emulsion was performed at the frequency range of 0.1–10 Hz to calculate the viscoelastic data of the emulsion.

### 3.11. Physical Stability of the Emulsions

The stability of the emulsions was analyzed by a dispersion analyzer LUMiSizer-651 (Berlin, Germany). The emulsion was loaded into a 2 mm PC centrifuge tube adapted to LUMiSizer-651, scanned at a wavelength of 870 nm and centrifuged at 3500 rpm for 100 min at 25 °C to plot 100 light transmittance curves. The instability index and interface tracking of emulsions were analyzed using SEPView6 software (LUM, Berlin, Germany).

### 3.12. Statistical Analysis

All data were statistically analyzed using SPSS-Statistics V17 software (IBM, Armonk, NY, USA) through the analysis of variance (ANOVA), followed by Duncan multiple comparisons [61]. The difference between the means was considered significant at *p* < 0.01.

## 4. Conclusions

The apparent viscosity and interfacial activity of two compound polysaccharides ((PGA + DOP) and (GA + DOP)) were studied, and oil-in-water emulsions were prepared using the two compound-polysaccharides as emulsifiers. The results showed that the two mixed polysaccharides had better interfacial activity compared to the individual polysaccharides, while the apparent viscosity of the two compound polysaccharides mainly depended upon the content of DOP. Similarly, the emulsions prepared with compound polysaccharides displayed smaller droplet sizes and better stability. Among all emulsions, the emulsions prepared by PGA:DOP = 2:8 (total concentration = 1.5 wt%) and those prepared by 2.0% GA + 1.5% DOP exhibited smaller average droplet sizes, better storage stability, and higher physical centrifugal stability. The results of this study confirmed our hypothesis, providing a system for compound-polysaccharides as emulsifiers. PGA contains numerous esterified carboxyl groups and good hydrophobicity, while DOP is a highly hydrophilic polysaccharide with almost no hydrophobic groups. Therefore, the compound polysaccharide of PGA and DOP is an amphiphilic mixture with good emulsifying properties. Furthermore, GA possesses high hydrophilicity and low viscosity; a stable emulsion requires a higher amount of GA, but the addition of DOP can solve this problem. The DOP could significantly increase the viscosity of the mixed polysaccharides of GA and DOP, and the increase in the viscosity of the continuous phase could significantly improve the stability of the emulsion. In addition, it is exciting that there may be interactions between compounded polysaccharide molecules. For example, the interfacial tension and apparent viscosity of compound polysaccharides are not a single change rule. Previous studies have demonstrated the interaction between different polysaccharide hydrocolloid molecules and the electrostatic interaction between polysaccharides and proteins [1,7,9,62,63,64,65,66]. Therefore, we have evidence to speculate that there is a molecular interaction between DOP and PGA or DOP and GA. This is exactly the research that we are going to do, which is to explore the interaction of DOP with other polysaccharides and their properties in aqueous solution. Ultimately, this study provides a valuable perspective on the application of natural plant polysaccharides as an emulsifier, and offers a new approach for the application of *Dendrobium officinale* polysaccharides.

## Figures and Tables

**Figure 1 molecules-25-00759-f001:**
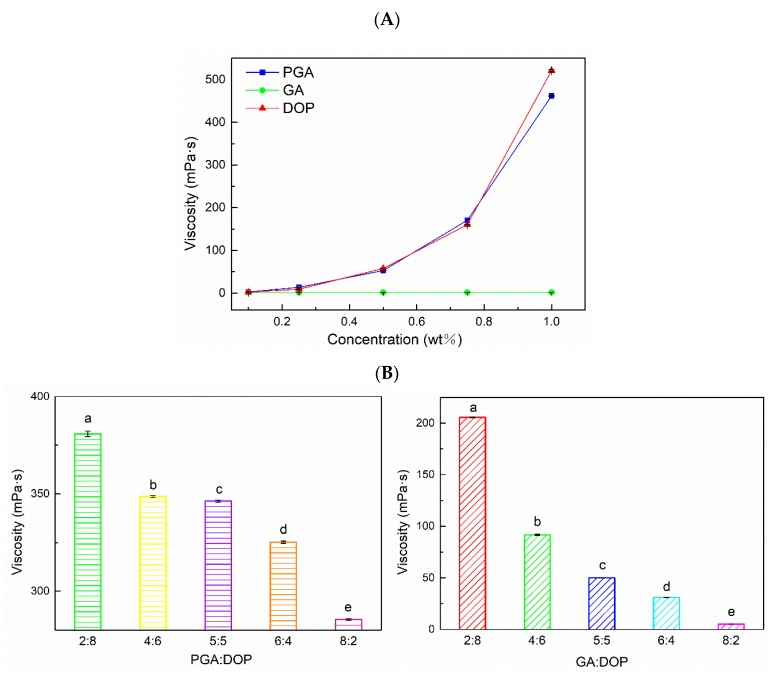
(**A**) Relationship between apparent viscosity and concentration of three polysaccharides. (**B**) Apparent viscosity of the compound polysaccharides at different ratios (total concentration was 1 wt%); a–e indicate significant differences (*p* < 0.01).

**Figure 2 molecules-25-00759-f002:**
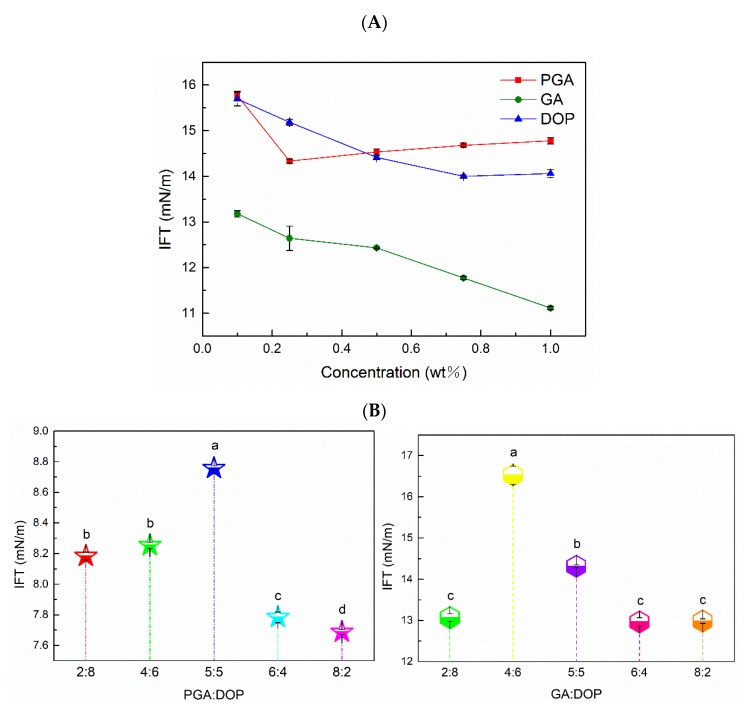
(**A**) Relationship between the interfacial tension (IFT) and concentration of three polysaccharides. (**B**) Interfacial tension (IFT) of the compound polysaccharides at different ratios (total concentration was 1 wt%), a–d indicate significant differences (*p* < 0.01).

**Figure 3 molecules-25-00759-f003:**
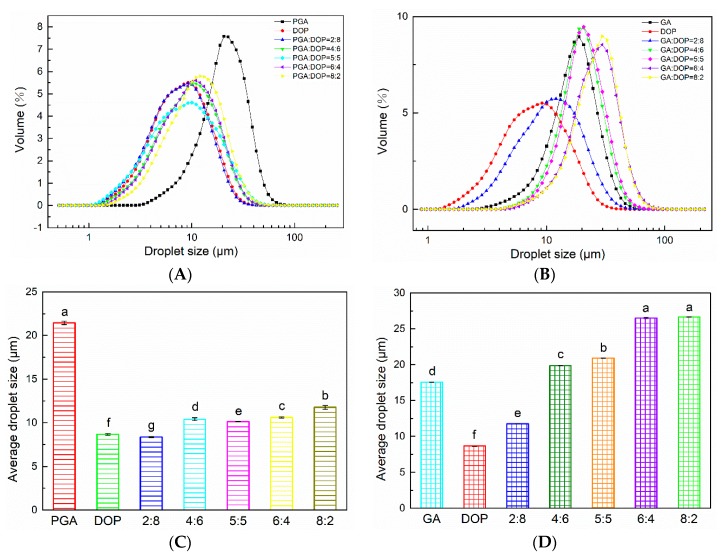
(**A**) Particle size distribution of emulsions prepared by compounding polysaccharides of PGA and DOP as emulsifiers (total emulsifier concentration was 1 wt%); (**B**) Particle size distribution of emulsions prepared by compounding polysaccharides of GA and DOP as emulsifiers (total emulsifier concentration was 1 wt%); (**C**) The average particle size of emulsions prepared by compounding polysaccharides of PGA and DOP as emulsifiers (total emulsifier concentration was 1 wt%); (**D**) The average particle size of emulsions prepared by compounding polysaccharides of GA and DOP as emulsifiers (total emulsifier concentration was 1 wt%), a–g indicate significant differences (*p* < 0.01).

**Figure 4 molecules-25-00759-f004:**
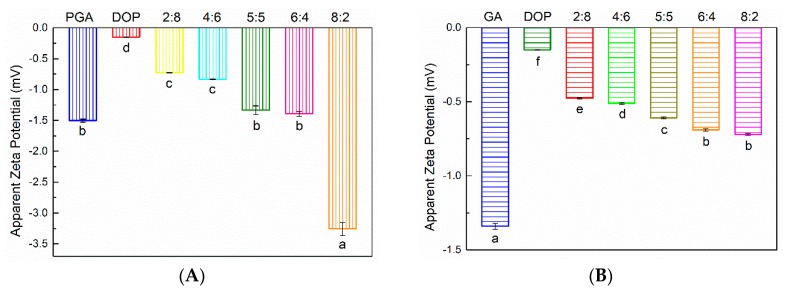
(**A**) The zeta potential of emulsions prepared by compounding polysaccharides of PGA and DOP as emulsifiers (total emulsifier concentration was 1 wt%); (**B**) The zeta potential of emulsions prepared by compounding polysaccharides of GA and DOP as emulsifiers (total emulsifier concentration was 1 wt%), a–f indicate significant differences (*p* < 0.01).

**Figure 5 molecules-25-00759-f005:**
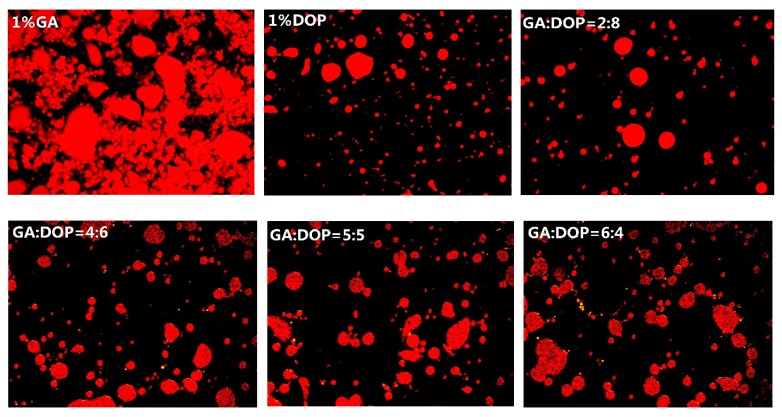
The microstructure of emulsions prepared by compounding polysaccharides of PGA and DOP or compounding polysaccharides of GA and DOP as emulsifiers (total emulsifier concentration was 1 wt%).

**Figure 6 molecules-25-00759-f006:**
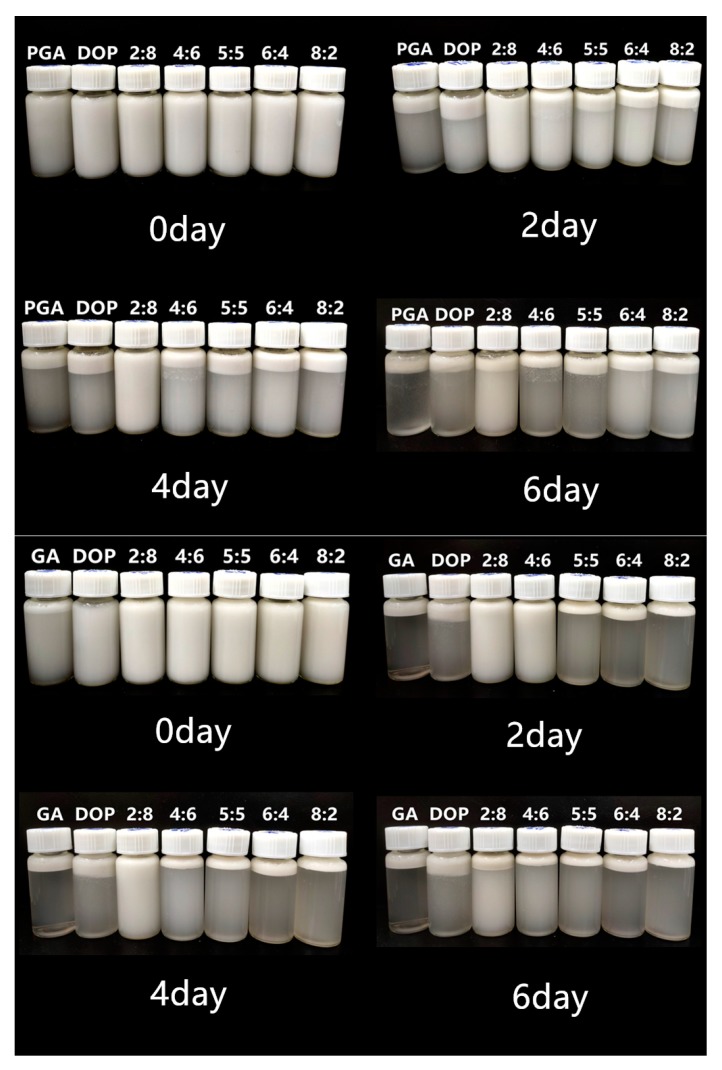
Visual observations (total emulsifier concentration was 1 wt%).

**Figure 7 molecules-25-00759-f007:**
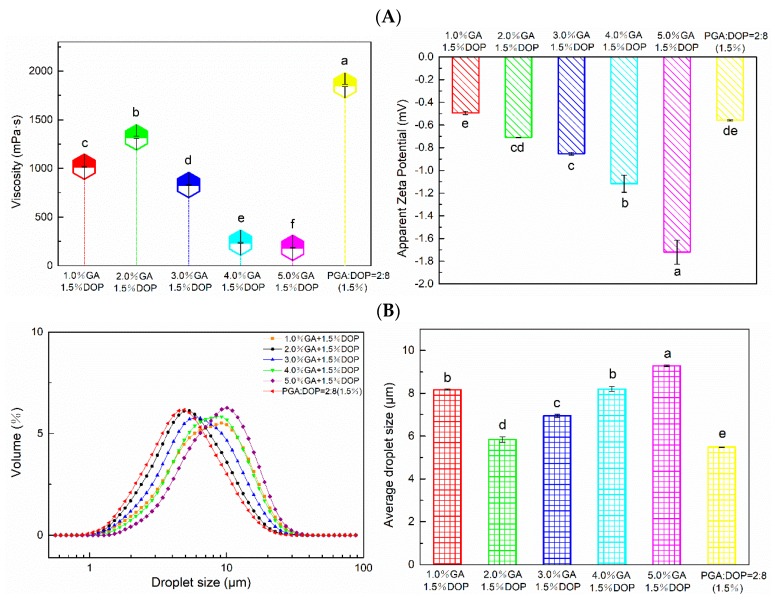
(**A**) The apparent viscosity and zeta potential. (**B**) The average particle size and particle size distribution. a–d, cd, e, de, f indicate significant differences (*p* < 0.01).

**Figure 8 molecules-25-00759-f008:**
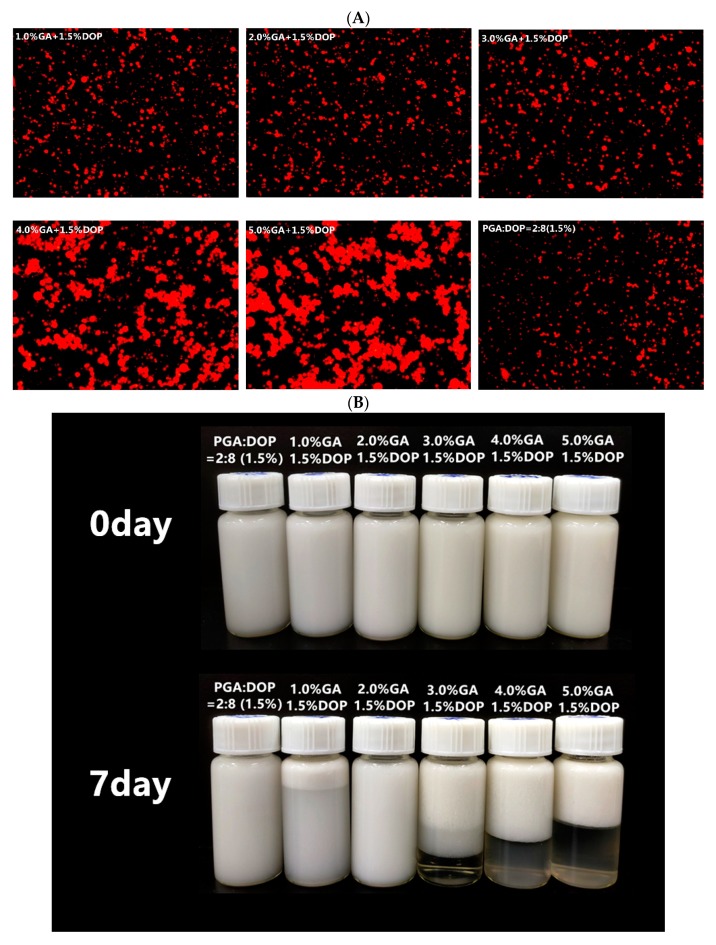
(**A**): The microstructure of emulsions (secondary screening of the concentration and ratio of compound polysaccharide emulsifiers); (**B**) The visual observations of emulsions (secondary screening of the concentration and ratio of compound polysaccharide emulsifiers).

**Figure 9 molecules-25-00759-f009:**
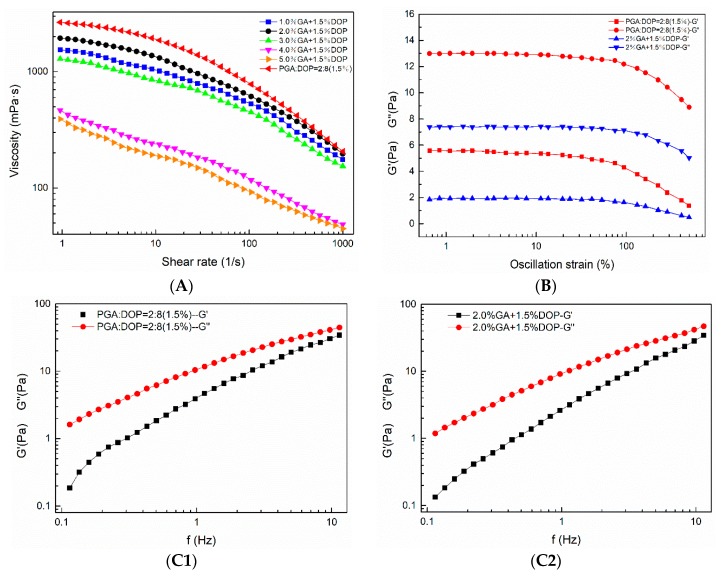
Rheological properties. (**A**) Steady-state flow curve. (**B**) Linear viscoelastic zone. (**C1**) Dynamic frequency sweep of the emulsion prepared by PGA: DOP=2:8 (1.5%); (**C2**) Dynamic frequency sweep of the emulsion prepared by 2.0% GA + 1.5% DOP.

**Figure 10 molecules-25-00759-f010:**
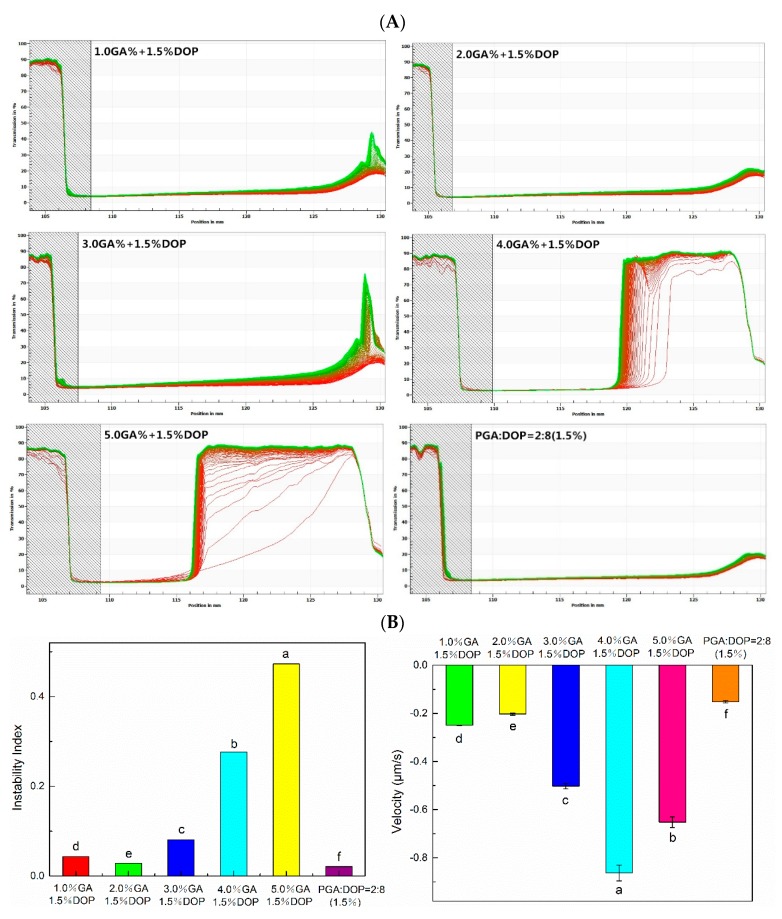
(**A**) Light transmittance curve. (**B**) Instability analysis and interface tracking analysis. The analysis range is from the meniscus to the bottom; a–f indicate significant differences (*p* < 0.01).

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
