# Peer review of "Stabilizing the Oil-in-Water Emulsions Using the Mixtures of Dendrobium Officinale Polysaccharides and Gum Arabic or Propylene Glycol Alginate"

_molecules, 2020, doi:10.3390/molecules25030759_

Round 1
Reviewer 1 Report
The paper describes the stabilizing effect of polysaccharides on the oil-in-water emulsions. The paper is correctly designed but should be improved before consideration of its publication.
The main remarks:
the presentation of the results should be more briefly the zeta potential value is quite high. It should be discussed Figures 1-4, 7, 10 - the literal indications (a-g) is not clear The discussion of the obtained results is poor. The authors should comper their results with the literature dataThe minor remarks:
is the authors' list completed? typographic errors connected with zeta potential according to Journal requirements the methods section should be at the end of the paperAuthor Response
Please see the attachment.

Reviewer 2 Report
1) It is not clear what the authors mean in their wording. There are two physically very different effects in emulsifying properties: a) the surface activity of emulsifiers, b) the increase of viscosity, and the connected shift of time scales, which prevents coalescence. Both have different molecular origin and need to be described in a journal called “Molecules”.
2) The structural details of the different and non-standard polysaccharides and modified alginates should be given in the introduction to provide at least an idea of understanding the result. Especially, when polysaccharides are used, which are not as well known as the standard thickeners.
3) as ha hint: The structure of the for the paper important polysaccharide DOP has been to some extend analysed in Ma, H., Zhang, K., Jiang, Q., Dai, D., Li, H., Bi, W., & Chen, D. D. Y. (2018). Characterization of plant polysaccharides from Dendrobium officinale by multiple chromatographic and mass spectrometric techniques. Journal of Chromatography A, 1547, 29-36.
The paper should really be rewritten to provide ideas beyond simple data presentation, as it is often the case in classical food technology journals. I encourage resubmission, if "molecular" interpredations and insights are possible. Alternatively, the authors may try submit their manuscript in more technological / applied journals.
Round 2
Reviewer 1 Report
The manuscript has been partially improved. It can be consider for publication.
Reviewer 2 Report
The manuscript reads now better and can be published.